# A High-Precision Quartz Resonant Ultra-High Pressure Sensor with Integrated Pressure Conversion Structure

**DOI:** 10.3390/mi14091657

**Published:** 2023-08-25

**Authors:** Quanwei Zhang, Cun Li, Huafeng Li, Yan Liu, Jue Wang, Xiaolong Wang, Yuan Wang, Fabin Cheng, Haijun Han, Peng Zhang

**Affiliations:** 1Institute of Systems Engineering, China Academy of Engineering Physics, Mianyang 621900, China; zhangqw1993@caep.cn (Q.Z.); lihuafeng72@163.com (H.L.); liuyan123@caep.cn (Y.L.); swcaep@163.com (J.W.); allersa@163.com (X.W.); orionwang@foxmail.com (Y.W.); cfb222@163.com (F.C.); hanhaijun@foxmail.com (H.H.); zhangp@caep.cn (P.Z.); 2The State Key Laboratory for Manufacturing System Engineering, Xi’an Jiaotong University, Xi’an 710049, China

**Keywords:** pressure sensors, quartz resonator, DETF, MEMS, ultra-high pressure

## Abstract

A quartz resonant pressure sensor is proposed for high-precision measurement of ultra-high pressure. The resonant unit realizes a push–pull differential layout, which restrains the common-mode interference factor, and the resonator is only subject to axial force. The pressure conversion unit is made in an integrated manner, avoiding output drift problems caused by residual stress and small gaps during assembly, welding, and other processes in sensor preparation. Theoretical and simulation analysis was conducted on the overall design scheme of the sensor in this paper, verifying the feasibility. Sensor prototypes were created and performance experiments were conducted. The experimental results show that the sensitivity of the ultra-high pressure sensor is 46.32 Hz/MPa at room temperature within the pressure range of 120 MPa, and the comprehensive accuracy is 0.0266%. The comprehensive accuracy of the sensor is better than 0.0288% FS in the full temperature range environment. This proves that the sensor scheme is suitable for high-precision and high-stability detection of ultra-high pressure, providing new solutions in special pressure measurement fields such as deep-sea and oil exploration.

## 1. Introduction

With the advent of the information age, sensor technology is becoming increasingly important and has become a strategic development field for countries around the world [1,2]. Pressure sensors are widely used in civilian fields such as automobiles and ships, consumer electronics, and military fields such as aerospace and weapon equipment [3,4]. With the development of application technology, especially in special application fields such as high-pressure measurement, there is a higher demand for the performance of pressure sensors.

At present, MEMS pressure sensors in the high-pressure special field mainly include three types: piezoresistive [5,6], fiber optic [7,8], and resonant [9,10]. Among them, MEMS piezoresistive pressure sensors are the mainstream products in the field of high-pressure measurement. The principle is to use the piezoresistive effect to make a sensor chip containing a sensitive resistance bridge circuit, which is usually packaged in a metal shell filled with silicone oil, and the corrugated diaphragm is used to convert the high pressure to be measured. Its advantages lie in its small size and low cost. MEMS piezoresistive pressure sensors need to consider the negative impact of assembly issues such as the chip itself, between the chip and the housing, and between the corrugated diaphragm housing, which will reduce the accuracy and stability of the sensor. A MEMS optical fiber pressure sensor uses the propagation characteristics of light in optical fiber materials to sense the external pressure to be measured on photoelasticity elements. Fiber optic pressure sensors are often used for ultra-high pressure measurement in various extreme environments, such as oil wells and seabed exploration, due to their suitability for harsh working conditions and strong electromagnetic interference environments. However, fiber optic pressure sensing also has its own limitations, as there are many factors that affect the variation of light intensity. Therefore, there are extremely high requirements for the precision of the sensor structure, the quality of the light source, and the assembly process, which limits the measurement accuracy.

The performance of MEMS resonant pressure sensors is influenced by the inherent characteristics of the resonant unit material and the overall mechanical structure. They exhibit advantages such as high accuracy, high stability, high anti-interference ability, and high resolution, and are widely used in fields that require high-precision pressure measurement. For example, Paroscientific has proposed a resonant pressure sensor product with a quartz vibrating beam as the resonant unit and a metal Bourdon tube as the pressure conversion unit [11]. Epson has developed a resonant pressure sensor product with a quartz tuning fork as the resonant unit, a metal corrugated tube as the pressure conversion unit, and a flexible rod [12]. The above products are widely used in the field of deep-sea exploration to measure sea depth and sea pressure and can also be used to predict or study tsunamis, tides, submarine volcanos, and other phenomena [13]. However, the existing pressure conversion units of such products have multiple structural assembly problems, which will inevitably bring negative factors such as residual stress and small gaps caused by assembly, welding, etc., in the preparation process, affecting the performance of the sensors. Therefore, existing products must undergo extensive compensation work to ensure their accuracy and stability.

In order to solve the above problems, this paper proposes an integrated sensor structure scheme for the field of large-scale and high-precision pressure measurement. In this scheme, quartz double ended tuning forks (DETFs) with piezoelectric excitation and piezoelectric detection are used. Quartz has the advantages of excellent mechanical properties, good elasticity, high bending strength, minimal hysteresis and creep, high quality factor, and good frequency stability. It is an ideal material for making high-precision and highly stable resonant units. Moreover, quartz also has excellent electrical properties, as their piezoelectric properties allow for simple excitation and detection of resonant units. Beryllium bronze has high hardness, elastic limit, fatigue limit, and wear resistance, as well as good corrosion resistance, thermal conductivity, conductivity, non-magnetism, fatigue resistance, and stress relaxation resistance. It is widely used as a material for making elastic components in various important fields. Moreover, the difference in thermal expansion coefficients between beryllium copper alloy and quartz single crystal is relatively small, reducing the adverse effects of temperature environment. The resonant unit realizes a push–pull differential layout, which curbs the common-mode interference factor, and the quartz resonator is only subject to the axial force, which solves the problem of uneven stress distribution on the two tines when the DETF is subjected to the width square bending moment, and improves the measurement accuracy of the sensor. The pressure conversion unit is made of beryllium copper material and processed in an integrated manner, achieving pressure measurement in high-pressure environments, avoiding the complex process of multi-structure assembly, solving the problem of output drift caused by residual stress and small gaps in assembly, welding, and other processes in sensor preparation, and improving the stability of the sensor.

## 2. Design

The ultra-high pressure quartz resonant sensor is generally divided into a resonant unit and a pressure conversion unit. The pressure to be measured is converted and transmitted to the resonant unit through the pressure conversion unit, and the output frequency of the resonant unit changes to complete the measurement.

### 2.1. Design of Resonant Unit

#### 2.1.1. Theoretical Analysis

The resonant unit adopts a DETF. When two tines of the DETF vibrate inversely in the same plane, the forces and moments acting on the roots of the two tines cancel each other, as shown in Figure 1, so that the fixed connection end of the DETF has less energy coupling with the outside world, less energy loss when resonance occurs, and high quality factor [14]. The two tines undergo reverse lateral bending vibration along the width direction in the same plane, and the two tines are in a completely symmetrical state on the same path. It is necessary to start from its own force–frequency characteristics, and to study the impact of its structural parameters on key performance indicators.

When an axial force *F* acts on the DETF, the force in both tines is *F*/2. Because the length of the tine is much greater than its width and thickness, the differential equation describing bending vibration can be obtained [15]:(1)$$\frac{Ew^{3}t}{12}\cdot \frac{\partial^{4}u}{\partial y^{4}}-\frac{F}{2}\cdot \frac{\partial^{2}u}{\partial y^{2}}-4\pi^{2}f^{2}\rho twu=0$$
where *u* is the displacement of the neutral plane of the tuning fork tines from the equilibrium position, *y* is the position coordinate along the length of the tuning fork tines, *F* is the axial force acting on both ends of the DETF, *f* is the resonant frequency during bending vibration, *ρ* is the density of the quartz crystal, *E* is the elastic modulus of the quartz crystal, and *w* and *t* are the width and thickness of the tuning fork tines, respectively.

The initial frequency *f*_0_ and the linear term force–frequency coefficient *S* are, respectively:(2)$$f_{0}=\frac{1.027w}{l^{2}}\sqrt{\frac{E}{\rho }}$$
(3)$$S=\frac{f-f_{0}}{f_{0}F}=\frac{0.0737l^{2}}{Etw^{3}}$$
where *l* is the length of the tuning fork tines.

The theoretical optimization analysis of the characteristic size of the DETF can obtain that within the selected range, the length and width of the tines must be large enough, which can greatly reduce the processing requirements. However, the longer the tine is, the smaller the fundamental frequency is, and the wider the tine is, the smaller the frequency sensitivity coefficient is. Therefore, it is also necessary to consider the impact of its feature size parameters on performance and balance the contradictory relationship between process difficulty and the performance of DETFs.

The quality factor is one of the important indicators of the resonant pressure sensor design scheme in this paper, which mainly affects the energy loss, frequency stability, and frequency selection ability of the resonant unit during vibration. The quality factor of resonator components is related to their geometric structure parameters, gas packaging environment, and other factors, mainly depending on various energy loss mechanisms. The main factors that affect the quality factor of quartz DETF are structural damping and gas damping. The structural damping losses of DETF resonators include thermoelastic losses, support losses, and surface energy losses. The gas damping of DETF mainly includes the air resistance damping of the resonant device, the sliding film damping of the upper and lower surfaces, and the squeeze film damping between the two tines [16].

#### 2.1.2. Simulation Analysis

In order to verify the above theoretical analysis and further optimize the characteristic size parameters of the DETF, the finite element software has been used for simulation optimization. The advantage of simulating and optimizing the characteristic dimensions of DETFs lies in not only considering the length and width of the tine calculated during theoretical analysis, but also incorporating the thickness and spacing of the tine into the optimization range, further comprehensively optimizing the resonant device. The modal simulation module has been used to simulate the working modal frequency and force–frequency sensitivity coefficient of DETF under different size parameters. Simulate and calculate the frequency of quartz DETFs with different feature sizes, and then calculate the frequency when a certain axial tension and pressure (preset as 1 N) are applied to the end of the DETFs to obtain the influence of feature size parameters on the force frequency sensitivity coefficient.

As shown in Figure 2, from the simulation results, it can be concluded that for the working fundamental frequency of quartz DETFs, the length of the tine has the greatest impact, followed by the width of the tine. The thickness and spacing of the tine have the smallest impact, and the influence of the spacing of the tine on the working fundamental frequency can be ignored. Regarding the resonant frequency of quartz DETFs subjected to tension force F at both ends, the width of the tine has the greatest impact (the variation range of the internal frequency sensitivity coefficient is greater than 0.3), followed by the length of the tine (the variation range of the internal frequency sensitivity coefficient is greater than 0.15), and the thickness of the tine has a small impact (the variation range of the internal frequency sensitivity coefficient is about 0.09). The influence of tine spacing on the forced vibration frequency is minimal (the variation range of the internal force frequency sensitivity coefficient is about 0.01) and can be ignored.

The simulation calculation results are consistent with the theoretical analysis results mentioned above, and both confirm the rationality and accuracy of each other, ensuring the effectiveness of the optimization content for the characteristic size of the quartz DETF resonator in this paper, and determining the design size, as shown in Table 1. The theoretical value of the quality factor of the DETF resonator in atmospheric air is 3214, and in vacuum it is 31,513 [16].

### 2.2. Design of Pressure Conversion Unit

A push–pull structure integrated sensor structure scheme has been proposed, as shown in Figure 3. The pressure conversion unit is composed of an integrated pressure circular film and a flexible push rod, which avoids the adverse effects of heterogeneous materials and split structures, and reduces the problem of random errors caused by residual stresses and small gaps during assembly and welding processes in the sensor structure. The resonant unit adopts a push–pull differential structure, which not only reduces conjugate interference, but also the DETF resonator is only subjected to axial force, avoiding the problem of uneven stress distribution on the two tuning fork tines caused by the bending moment in the width direction of the DETF, and improving the measurement accuracy of the sensor.

#### 2.2.1. Theoretical Analysis

After determining the overall design of the push–pull integrated structure scheme, theoretical analysis and establishment of a theoretical calculation model are carried out. Its operating mechanism is that the pressure to be measured is converted through a pressure circular membrane and transmitted to the flexible push rod, so that the resonant unit of the differential structure senses the mechanical signal. The simplified structure diagram is shown in Figure 4. The simplified mechanical model can be divided into two parts, one part being a flexible push rod (Point A) and the other part being a pressure circular membrane (Point B).

The pressure circular membrane converts *F_P_* into *F_G_*, which is then converted into *F_OUT_* through a square rod flexible push rod and applied to the resonant unit. According to the mechanical equilibrium equation and deformation compatibility equation, we can get the mechanical equations of the push–pull structure integrated structure scheme:(4)$$\left\{ \begin{array}{l} F_{M}=k_{M}\delta_{A}\\ F_{G}=k_{G}\delta_{G}\\ F_{TL}=k_{TL}\delta_{B}\\ F_{OUT}=k_{OUT}\delta_{B}\\ F_{P}=F_{M}+F_{G}\\ F_{G}=F_{TL}+F_{OUT}\\ \delta_{A}-\delta_{G}=\delta_{B} \end{array}\right.$$

In the formula, *F_P_* is the force exerted by the pressure to be measured in the vertical direction of the pressure circular membrane; *F_M_* is the reaction force when the elastic deformation of the pressure circular membrane occurs; *k_M_* is the deformation stiffness of the pressure circular membrane in the vertical direction; *δ_A_* is the deformation that occurs in the vertical direction at the center point A of the pressure circular membrane; *F_G_* is the reaction force when a square rod undergoes tensile elastic deformation; *k_G_* is the tensile and compressive stiffness of a square rod; *δ_G_* is the axial compression deformation of the square rod; *F_TL_* is the reaction force when the flexible push rod undergoes elastic deformation; *k_TL_* is the deformation stiffness of the flexible push rod when elastic deformation occurs; *δ_B_* is the deformation of the center point B of the flexible push rod in the vertical direction; *F_OUT_* is the reaction force of the resonant unit on the flexible lever; *k_OUT_* is the tensile and compressive stiffness of the resonant unit.

By solving Equation (4), the force transmitted by the push–pull integrated structural scheme to the resonant element can be obtained:(5)$$F_{OUT}=\frac{k_{OUT}k_{G}}{k_{OUT}k_{G}+k_{M}(k_{OUT}+k_{G}+k_{TL})+k_{TL}k_{G}}F_{P}$$

As one of the components of the pressure conversion unit, the design of the structural size of the pressure circular membrane determines the sensitivity, and other performance of the sensor in this scheme. The deformation of the pressure circular membrane can be approximated as small deflection deformation, and the size of small deflection deformation of the pressure circular membrane can be calculated from the perspective of energy. Moreover, due to the presence of square rods, the pressure circular membrane of the push–pull integrated structure scheme cannot be simply regarded as a circular membrane. The connection between the pressure circular membrane and the square rod does not undergo bending deformation, and it needs to be considered as a circular membrane with a convex structure on the surface. At this point, the pressure to be measured acts simultaneously on the pressure circular membrane and its boss, and the deformation stiffness of the circular membrane with a boss structure in its vertical direction can be obtained:(6)$$k_{M}=\frac{3\left(1-\nu^{2}\right)R^{2}}{16\pi Ed^{3}}\left[1-{\left(\frac{w_{0}}{R}\right)}^{4}+4{\left(\frac{w_{0}}{R}\right)}^{2}\ln\left(\frac{w_{0}}{R}\right)\right]$$

In the equation, *ν* and *E* are Poisson’s ratio and elastic modulus of pressure circular membrane material (beryllium copper alloy), respectively; *R* is the radius of the pressure circular membrane; *w*_0_ is the side length of the square convex surface; *d* is the thickness of the pressure circular membrane.

The flexible push rod is the key structure of the pressure conversion unit in the integrated push–pull structure scheme and is the last step in transmitting mechanical signals to the resonant unit. Therefore, the design of its size and structural parameters will greatly affect the overall performance of the sensor in this scheme. The left and right parts of the flexible push rod are symmetrical structures, as shown in the figure, and their symmetrical structure can be simplified as a series structure composed of three rods with lengths of *l*_1_, *l*_2_, and *l*_3_. The bending stiffness of the flexible push rod for the reaction force *F_G_* of the square rod is equivalent to the stiffness of the parallel structure of the left and right parts, so the equivalent bending stiffness of the flexible push rod can be obtained:(7)$$k_{TL}=2\frac{k_{1}k_{2}k_{3}}{k_{1}+k_{2}+k_{3}},k_{1}=k_{2}=k_{3}=-\frac{l_{i}^{3}}{3EI_{i}},\;I_{i}=\frac{t_{i}w_{i}^{3}}{12}$$
where *k*_1_, *k*_2_, and *k*_3_ are the bending stiffness of three simplified series rods; *l_i_* represents the length of three simplified series rods (*i* = 1–3, *l*_1_, *l*_2_, *l*_3_); *I_i_* is the moment of inertia for the axial bending deformation of three simplified series rods (*i* = 1–3); *t_i_* represents the thickness of the flexible push rod; and *w_i_* represents the width of three simplified series rods (*i* = 1–3).

The function of the square rod in the integrated push–pull structure scheme is to connect the pressure circular membrane and the flexible push–pull structure. Under the action of the pressure to be measured, the square rod undergoes compression deformation, and the mechanical signal converted by the pressure circular membrane is transmitted to the flexible push rod structure. The expression for the axial tensile and compressive stiffness of the square rod is:(8)$$k_{G}=EA_{G}$$
where *A_G_* is the square cross-sectional area of the square rod.

For the tension and compression stiffness of the output resonant unit, as mentioned earlier, since the resonant unit is a differential output structure composed of two quartz DETFs, and its main deformation part is the tuning fork tine, that is, the stiffness of the resonant unit depends on the four parallel tuning fork tines, the tension and compression stiffness of the resonant unit can be obtained:(9)$$k_{\mathrm{OUT}}=4\frac{E_{q}A_{q}}{l_{q}}$$
where *E_q_* is the elastic modulus of quartz crystal; *A_q_* = *b_q_h_q_* is the cross-sectional area of the tuning fork tine; *l_q_* is the length of the deformed tuning fork tine of the resonant unit.

#### 2.2.2. Simulation Analysis

After establishing the theoretical model, simulation analysis, optimization, and validation have been conducted on the key feature size parameters obtained from the above analysis. The analysis of the sensor structure design uses the solid mechanics module, mainly simulating the force acting on the sensor pressure conversion unit and resonant unit under the pressure to be measured. According to the key dimensional parameters obtained, the full parameter scanning combination is enabled, and the solutions of all optimization conditions are simulated and calculated. Then, the axial stress distribution of the central axis of the resonant unit in this scheme can be obtained. After comprehensively considering the stress distribution along the y-axis direction of the resonant unit center axis and the maximum allowable stress limit in the overall simulation model of the sensor, the optimization solution was selected from the simulation optimization results within the full parameter scanning range, and the optimization of the structural size parameters of this scheme was completed. After confirming the optimized structural size parameters of the push–pull integrated structure scheme, the stress distribution and overall deformation of the sensor model in this scheme were simulated and analyzed.

As shown in Figure 5, the full range pressure to be measured causes deformation of the pressure circular membrane. The flexible push rod is pushed through a square rod to transmit the mechanical signal to the resonant unit of the differential output. It can be seen that when the sensor model is subjected to full range pressure, the maximum stress concentration is located at the pressure circular membrane, and the maximum stress is 294 MPa, which is far less than the elastic limit of beryllium copper material, which verifies the reliability of the overall sensor design of this scheme. As shown in Figure 6, under the action of the pressure to be measured, the integrated structure scheme of the push–pull structure has two quartz DETFs of the resonant unit, one stretched and the other compressed, with the same stress magnitude and opposite direction. Moreover, the stress in the y-direction along the tuning fork tine axis is the highest, while the stress in the other two directions is close to zero. At this point, the stress magnitude of the two quartz DETFs along the axial direction of the tuning fork tine is about ±45 MPa. The calculated frequency output after differential output is 5905.4 Hz, and the simulation sensitivity of the sensor is 49.21 Hz/MPa.

By substituting the optimization results, the theoretical value of the differential output of the sensor at full scale (120 MPa) can be calculated to be 6103.6 Hz, with a sensitivity of 50.86 Hz/MPa. Comparing the theoretical model with the simulation results, it can be seen that the theoretical calculation values of the performance parameters of the optimization scheme are basically consistent with the simulation verification results. The reason for the error is that the theoretical model has been simplified to some extent. For example, the symmetrical structure of the flexible push rod has been simplified to a series structure composed of three rods, and the theoretical expression of the pressure circular membrane also has assumptions. However, the error between the theory of the optimization scheme and the conclusion of the simulation model is relatively small, and the effectiveness of this scheme has been verified.

## 3. Fabrication

The production of MEMS quartz resonant pressure sensors in this paper mainly includes three aspects: firstly, the processing of resonant units, mainly including external dimension processing technology and surface electrode production; the second is the production of pressure conversion units, and a process plan is developed for the overall structure design of this paper; the third is the packaging scheme design of sensors.

### 3.1. DETF Resonator Unit

In summary, the shape of the quartz DETF resonator structure was completed through wet etching, and the surface electrodes are made by sputtering [17]. In order to solve the production problem of side electrodes, a process scheme combining mechanical hard mask and inclined sputtering was adopted. Using magnetron sputtering equipment, the excited metal atoms move in a straight line in a high vacuum sputtering chamber, tilting the slide table at a certain angle to allow the metal atoms to pass through the hollow area of the mechanical hard mask and deposit on the side of the DETF to form a specific pattern. The process flow is shown in Figure 7.

After completing the processing of the DETFs, the physical photo of the quartz DETF designed in this paper is shown in Figure 8. It can be seen that the shape of the resonator is intact, and the external dimensions are consistent with the design. By taking local photos of the decoupled area, tuning fork tine, and four sided electrodes under an optical microscope, it can be seen that the contour is clear, the electrode quality is good, and the sidewall electrode sputtering is successful, with obvious segmentation of the electrode strip. Through SEM local micrographs, the upper surface morphology of the tine is good, with clear electrode lines and high quality. It can be seen that the side wall is steep and without obvious crystal edges, and the segmentation of the side electrode is obvious, with good coverage. The process flow developed in this paper is reasonable and meets the design requirements. The preparation of a quartz DETF resonator has been completed.

### 3.2. Pressure Conversion Unit

Beryllium copper (QBe1.7), which is thermally matched with quartz crystal, was selected as the main elastic material of the pressure conversion unit. It has high hardness, elastic limit, fatigue limit, and wear resistance, as well as good corrosion resistance, thermal conductivity and conductivity, non-magnetism, fatigue resistance, and stress relaxation resistance. The integrated push–pull structure scheme has all structures made of beryllium copper material, as shown in Figure 9. The external structure of the elastic components in the resonant pressure sensor scheme adopts the electric discharge process. After the secondary parts (external threads, hexagonal surfaces, etc.) are machined, the external structure of the key parts is completed using the wire cutting process. The positioning groove and overflow groove of the quartz DETF of the resonant unit are made by the electric discharge etching process. In addition, the packaging shell structure is made using traditional turning mechanical processing technology, and all structures are annealed after processing to reduce their residual stress.

The natural frequency of the quartz DETF designed in this paper belongs to the low-frequency range (about 35 kHz), so the excitation circuit is designed using a logic double gate oscillator circuit. The resonant unit of the resonant pressure sensor in this paper is a differential structure, consisting of two DETFs. Therefore, the circuit requires two excitation detection circuits for excitation and frequency pickup, respectively. The PCB layout of the excitation detection circuit design is shown in Figure 10. The processed excitation circuit has undergone protection treatments and aging screening on the components and circuit board to ensure its reliability.

### 3.3. Sensor Packaging

According to the design scheme in this paper, a special quartz patch adhesive that is thermally matched with quartz crystals and beryllium copper was used, and its main component is epoxy resin adhesive doped with a specific proportion of quartz powder. The specially made quartz adhesive has good adhesion performance, certain fluidity, good impedance, and good insulation performance, avoiding thermal stress and residual stress during the curing process of the adhesive, and improving the stability of quartz DETF vibration. The curing method of stepwise heating and cooling (0–190 °C) was adopted to ensure the reliability of bonding and effectively reduce residual stress, as shown in Figure 11.

This paper adopts an integrated packaging and upper and lower chamber isolation scheme, as shown in Figure 12. Among them, the upper sensitive chamber is the chamber where the pressure conversion unit and resonance unit are located, the lower circuit installation chamber is the chamber where the PCB circuit board of the excitation detection circuit is located and completes the connection with external signals. The electrical connection between the two isolated chambers is achieved through four glass sintered lead pillars on the shell.

The seal gas composition and vacuum degree of the sensitive chamber have a greater impact on the quality factor of the resonant unit and the heat dissipation of the sensitive structure. In order to achieve thermal equilibrium when the quartz DETF vibrates, it is necessary to select the gas with higher thermal conductivity. Secondly, the vacuum degree should not be too high. the higher the vacuum degree is, the higher the quality factor will be. However, if the vacuum degree is too high, nonlinear vibration will occur, which is not conducive to vibration stability. Moreover, excessive vacuum is difficult to maintain for a long time, leading to long-term leakage and affecting the consistency of the packaging environment. Therefore, this sensitive chamber uses atmospheric dry air as the packaging gas, ensuring the heat dissipation of the resonator and the stability of the packaging environment.

There is only a small amount of specially made quartz adhesive present in the sensitive chamber, ensuring the moisture removal effect inside the chamber and facilitating the isolation and encapsulation of air tightness. The lower circuit installation chamber reduces the electromagnetic interference of the excitation detection circuit on the quartz DETF, which is conducive to the heat dissipation of the circuit board. The three prevention measures and the treatment of epoxy resin adhesive improve the reliability of the circuit. The entire process of sensor assembly and packaging in this paper has been completed in a clean laboratory. The actual sensor produced is shown in Figure 13. The total length of the prototype is 100 mm, and the diameter of the cylindrical end face is 50 mm.

## 4. Experiment and Discussion

The performance experiment of resonant pressure sensors in this paper mainly focuses on their static performance indicators, including sensitivity, fitting error, hysteresis error, and repeatability error. The environmental conditions for the performance experiment are standard atmospheric pressure, room temperature environment, relative humidity below 85%, and no external interference such as vibration, impact, or acceleration. In order to test the basic performance parameters of the sensor in this paper, it is necessary to build an experiment platform to conduct performance calibration experiments on the sensor prototype. The entire testing process is completed in a clean laboratory.

The experimental platform is built from a ground isolated battery, a stable power supply, a piston pressure control console, a frequency meter, a laptop, and a high and low temperature oven. The experimental principle of performance testing for resonant pressure sensors in this paper is shown in Figure 14. Among them, the ground isolation battery is used to provide electrical energy for all experimental testing equipment, prevent cross talk of laboratory power grounding wires, and reduce signal noise. The regulated power supply provides 5 V regulated DC for the excitation detection circuit of the sensor. The piston type pressure controller provides calibration pressure for the sensor prototype, which provides a controlled pressure output range of 0–250 MP and a control accuracy of 0.01 level. The frequency meter detects the two differential frequency signals output by the sensor during the experiment. The laptop is used as upper computers to read and save data collected by frequency meters. The high and low temperature oven provides a constant or variable temperature environment for the sensor.

### 4.1. Time Drift

The prerequisite for the zero time drift experiment of a pressure sensor is a constant (temperature, humidity, etc.) environment and no external signal input, which is an important indicator of the stability of the performance of the pressure sensor. Put the sensor prototype in the oven with a constant ambient temperature of 25 °C, supply power to the sensor prototype, preheat it for 1 h to make it reach thermal equilibrium, use the frequency meter to automatically collect and PC upper computer to record the frequency signal of the sensor prototype zero output, the sampling time is 6 h, and the sampling frequency is 250 ms.

The zero time drift curve of the output frequency of the sensor resonant unit is shown in Figure 15. It can be seen that the two zero output frequencies of the two resonators (DETF 1 and DETF 2) in the resonant unit have very small amplitude changes with time. In the overall curve of the two outputs, it can be seen as parallel straight lines with slight fluctuations. As shown in the figure, the two outputs of the resonant unit fluctuate around 33,679.4 Hz and 33,492.8 Hz, respectively; The maximum fluctuation range of DETF1 is about 0.13 Hz, and the overall change direction is slightly increased. The maximum fluctuation range of DETF2 is about 0.07 Hz, and its overall variation direction is slightly increasing.

The zero time drift variation of the differential output frequency signal of the sensor prototype is shown in Figure 16. It can be seen that the zero time drift of the sensor is stable, and the overall fluctuation range and amplitude of the change are small. In the 6 h zero time drift experiment of the sensor prototype, the overall fluctuation range of its differential output frequency is less than 0.25 Hz, and the calculated zero time drift is 0.0022% FS/h. The overall trend of the zero drift of the sensor prototype is slightly reduced, and the distribution of fluctuating data points is continuous and smooth. Although there are some isolated jump points (the change range of jump points is less than 0.15 Hz), the changes between adjacent two points are very small. This phenomenon indicates that the pressure conversion unit of the push–pull structure integrated scheme is achieved through integrated processing, ensuring the stability of the sensor structure, greatly reducing random errors, and improving the overall performance of the sensor.

### 4.2. Temperature Drift

In order to characterize the temperature stability of the sensors, zero temperature drift experiments were conducted on the sensor prototype. Place the prototype in the oven to supply power to the sensor prototype, use the frequency meter to automatically collect and record the frequency signal of the sensor prototype zero output on the PC upper computer; the test temperature range is −20–60 °C, the temperature change interval of the oven is 20 °C, the stability time of each test temperature point is 1 h, and the temperature change time is automatically controlled by the oven to complete a temperature cycling.

The zero position of the two output frequencies of the sensor prototype as a function of temperature was obtained. It can be seen that the zero temperature drift of the resonator output frequency of the resonant unit in the two schemes is approximately linearly increasing. As shown in Figure 17a, the slope of the linear growth change of the zero output of the two outputs of the resonant unit is approximately the same. This indicates that although the metal structure of the push–pull structure integrated scheme, especially the pressure conversion unit, undergoes thermal deformation with temperature in the zero temperature drift experiment, which leads to uneven force distribution between the two resonators, the difference between the two is small.

As shown in Figure 17b, the zero temperature drift of the differential output frequency of the sensor prototype shows an approximate linear increasing trend. This is because the slope of the increase in DETF1 of the sensor prototype is greater than that of DETF2, and the zero position of the differential output frequency gradually increases. The “opening” of temperature cycling is small, and the curve of temperature positive stroke and reverse stroke is consistent, which proves that the integrated push–pull structure has good thermal stability. The reference temperature point is set to 20 °C, and the calculated zero temperature drift value of the sensor prototype is 0.0194% FS/°C.

### 4.3. Comprehensive Accuracy

According to the comprehensive accuracy of the sensor prototype in this paper, the static pressure calibration experiment was carried out.

#### 4.3.1. At Room Temperature

The prototype was placed in a temperature oven, the oven temperature was constant at 25 °C, a 5 V constant voltage DC source was used to supply power to the sensor prototype, the atmospheric pressure of the clean laboratory was about 96.12 kPa, and a piston pressure controller was used to provide the pressure signal to be measured; the test range was 0–120 MPa, and a test point was set every 20 MPa within the measurement range. Starting from 0 MPa, positive stroke pressure loading was carried out, and after reaching the full range, a reverse stroke test of pressure unloading was carried out until zero point, and the output data were recorded at the test point during the whole process. Three forward and reverse stroke cycles of loading and unloading were performed.

The average value of three cyclic tests on the differential output frequency of the push–pull structure integrated sensor prototype increases from −186.99 Hz to 5371.02 Hz, resulting in a zero output value of −186.99 Hz and a full scale output value of 5558.01 Hz, which is approximately 16.6% of the resonant frequency of the resonator. Its sensitivity reaches 46.32 Hz/MPa, as shown in Figure 18.

Using polynomial fitting, the maximum deviation value reaches 1.26 Hz, and the compliance error is approximately 0.0227% FS. According to the hysteresis error calculation formula, the maximum deviation between the average values of the forward and reverse strokes is 0.62 Hz, and the hysteresis error of the sensor prototype is 0.011% FS. According to the repeatability calculation formula, the sub sample deviation within the entire measurement range is calculated to be 0.187 Hz. If a confidence coefficient of 2.5 is taken, the repeatability error of the sensor prototype is 0.0084% FS. In summary, the comprehensive accuracy of the sensor prototype in this paper is 0.0266% at room temperature. According to the theoretical and simulation calculations of the sensitivity of the sensor proposed in this paper, compared with the experimental results, as shown in Figure 19, the theoretical value of the sensor sensitivity is 48.09 Hz/MPa, and the simulation value is 44.18 Hz/MPa. The relative errors between the theoretical and simulation results and the experimental results are 6.24% and 9.80%, respectively. The comparative study verifies the feasibility of the proposed sensor prototypes in this paper and proves the rationality of the theoretical analysis and simulation calculations in this paper.

#### 4.3.2. Full Temperature Range

Conduct a full temperature range static pressure calibration experiment on the sensor prototype, keeping the oven temperature constant at −20–60 °C, with an interval of 20 °C as a node. Other test conditions are the same as the calibration experiment at room temperature. The experimental results of the full temperature range static calibration of the sensor prototype in this paper are shown in Figure 20.

Under environmental conditions of 0 °C, the full-scale output value of the differential output frequency of the push–pull structure integrated sensor prototype in three cycles of testing is 5560.16 Hz, and its sensitivity reaches 46.33 Hz/MPa. The maximum deviation value of polynomial fitting reaches 1.446 Hz, and the compliance error at this time is approximately 0.026% FS. According to the hysteresis error calculation formula, the maximum deviation between the average values of forward and reverse strokes is 0.556 Hz, and the hysteresis error of the sensor prototype is 0.01% FS. According to the repeatability calculation formula, the sub sample deviation within the entire measurement range is calculated to be 0.14 Hz. If a confidence coefficient of 2.5 is taken, the repeatability error of the sensor prototype is 0.0063% FS. The calculated comprehensive accuracy is 0.0286% FS. Similarly, the comprehensive accuracy of the sensor prototype under other temperature conditions has been calculated, as shown in Table 2. It can be seen that the comprehensive accuracy of the sensor prototype is better than 0.0288% FS in the full temperature range environment, which meets the requirements of high-precision pressure measurement.

## 5. Conclusions

A quartz resonant pressure sensor is proposed for high-precision measurement of ultra-high pressure. The resonant unit realizes a push–pull differential layout, which restrains the common-mode interference factor, and the resonator is only subject to axial force. The pressure conversion unit is made in an integrated manner, avoiding output drift problems caused by residual stress and small gaps during assembly, welding, and other processes in sensor preparation. Theoretical and simulation analysis has been conducted on the overall design scheme, verifying the feasibility of the design scheme. The experimental results show that within the pressure range of 120 MPa at room temperature, the sensitivity of the ultra-high pressure sensor is 46.32 Hz/MPa, the compliance error is about 0.0227% FS, the hysteresis error is 0.011% FS, and the repeatability error is 0.0084% FS. The comprehensive accuracy of the sensor prototype at room temperature is 0.0266%. In addition, sensor performance experiments have been conducted in a full temperature range environment, and the results show that the comprehensive accuracy is better than 0.0288% FS. This proves that the sensor in this paper can meet the requirements of ultra-high pressure measurement in special fields and has great application potential, providing a new solution in special pressure measurement fields such as deep-sea and oil exploration.

## Figures and Tables

**Figure 1 micromachines-14-01657-f001:**
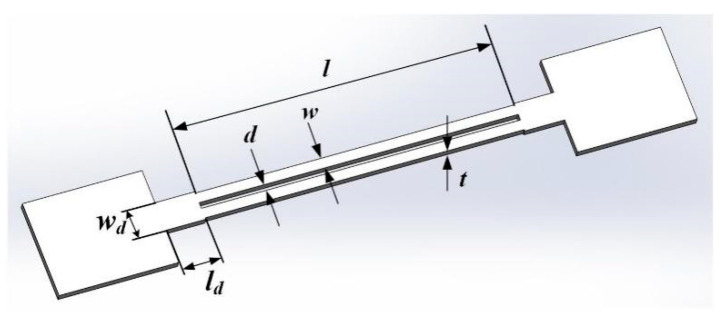
Schematic diagram of the structural shape and characteristic dimensions of a DETF.

**Figure 2 micromachines-14-01657-f002:**
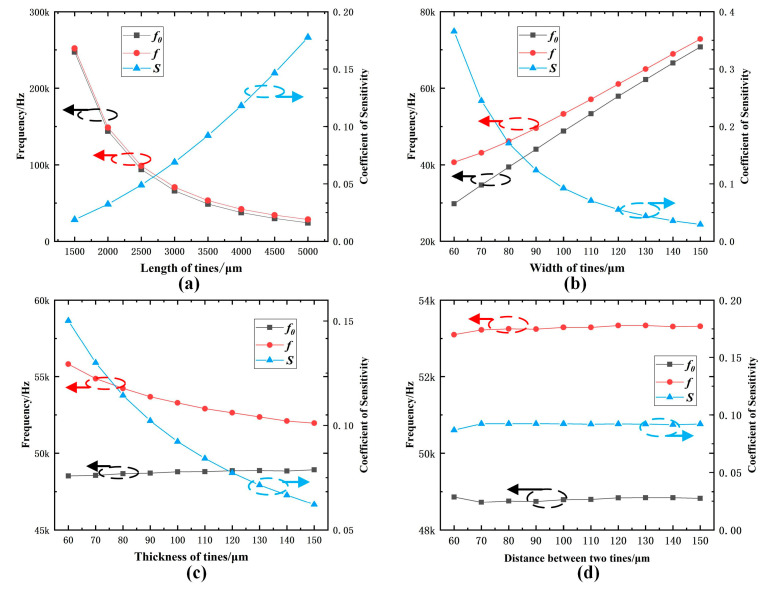
Schematic diagram of the structural shape and characteristic dimensions of a DETF: (**a**) Change in length; (**b**) Change in width; (**c**) Change in thickness; (**d**) Change in distance between two tines.

**Figure 3 micromachines-14-01657-f003:**
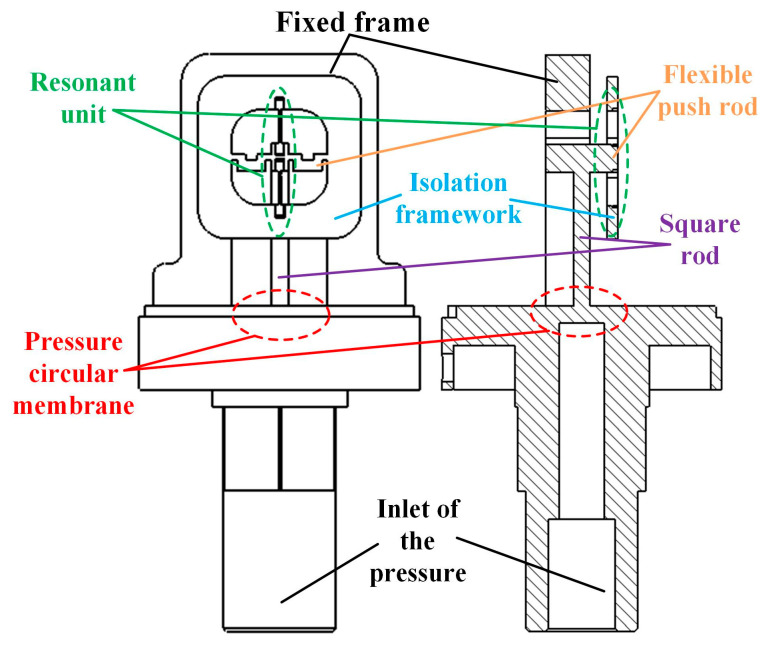
Schematic diagram of the overall structure of the push–pull integrated structure.

**Figure 4 micromachines-14-01657-f004:**
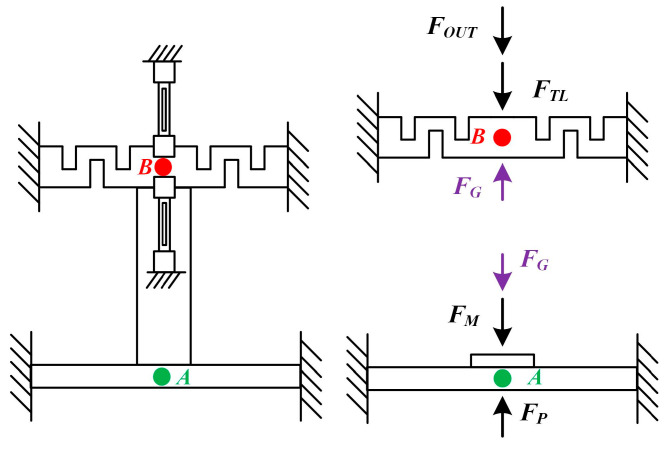
Mechanical analysis diagram of the sensor structure.

**Figure 5 micromachines-14-01657-f005:**
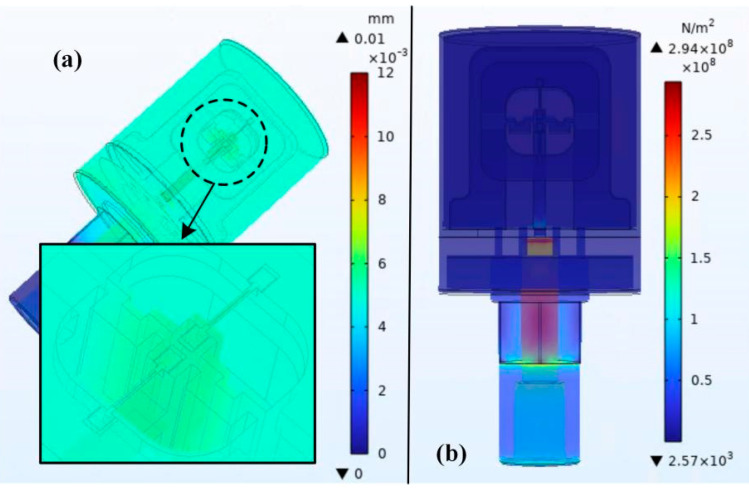
Full-scale simulation of push-pull integrated structure optimization scheme: (**a**) Displacement; (**b**) Strain.

**Figure 6 micromachines-14-01657-f006:**
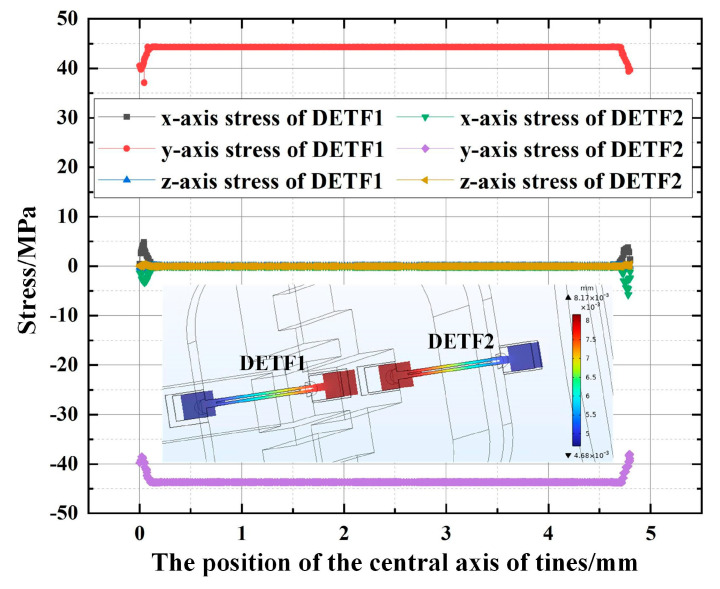
Full-scale simulation of integrated structure optimization scheme of push-pull structure stress distribution on the central axis of tuning fork tines.

**Figure 7 micromachines-14-01657-f007:**
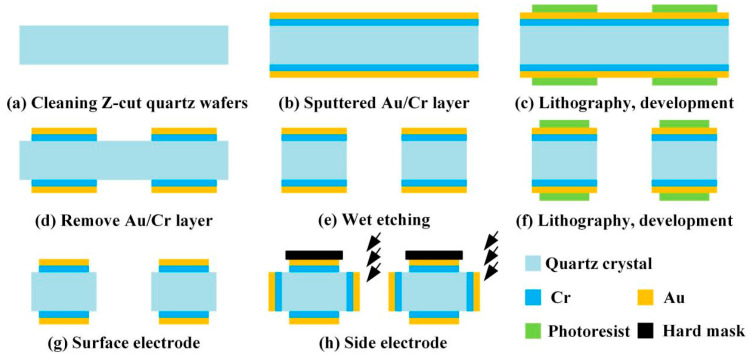
Processing process of quartz DETFs: (**a**) Cleaning; (**b**) Sputtered Au/Cr layer; (**c**) Lithography and development; (**d**) Remove Au/Cr layer; (**e**) Wet etching; (**f**) Lithography and development; (**g**) Surface electrode; (**h**) Side electrode.

**Figure 8 micromachines-14-01657-f008:**
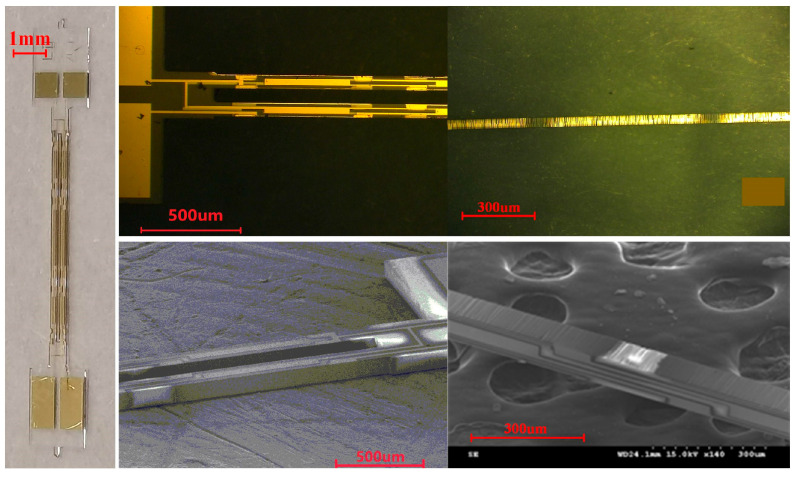
Photos of the completed quartz DETF.

**Figure 9 micromachines-14-01657-f009:**
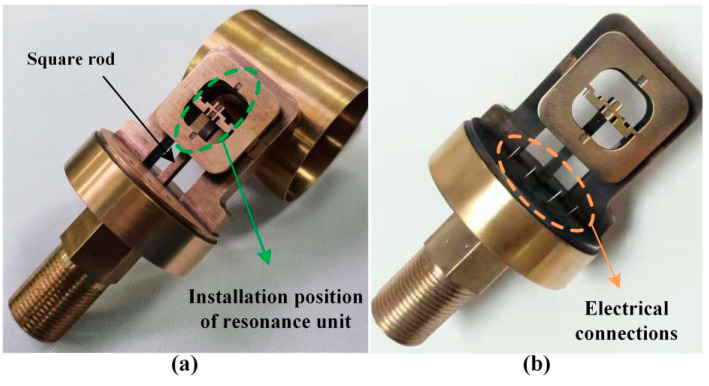
Physical diagram of sensor pressure conversion unit and cover: (**a**) Structure introduction; (**b**) Electrical connection location.

**Figure 10 micromachines-14-01657-f010:**
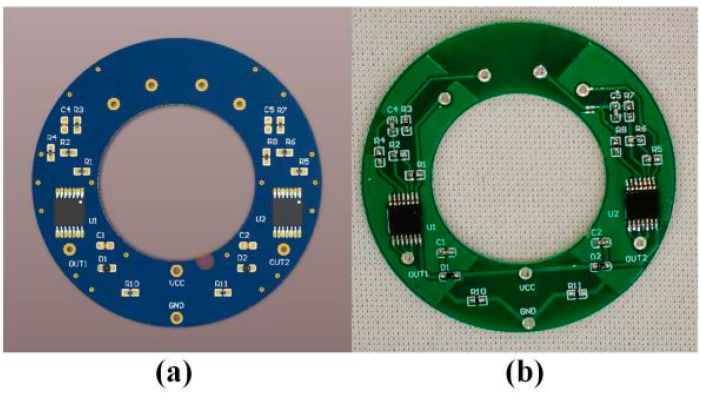
Excitation detection circuit: (**a**) Design drawings; (**b**) Physical image.

**Figure 11 micromachines-14-01657-f011:**
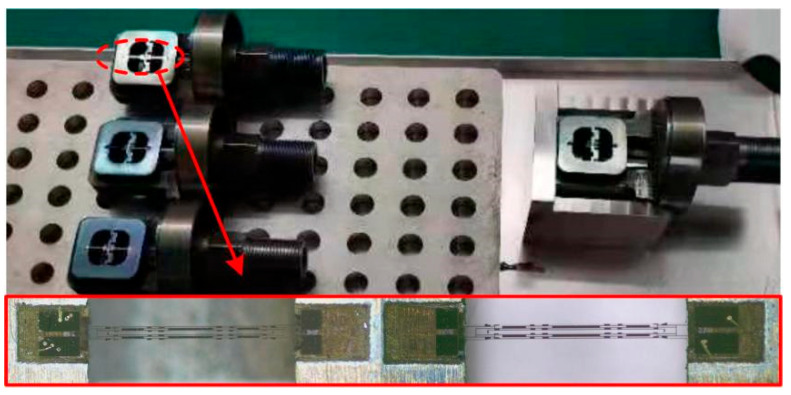
Quartz DETF mounting process.

**Figure 12 micromachines-14-01657-f012:**
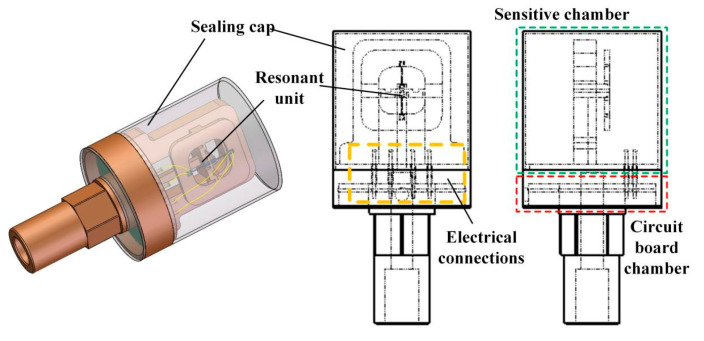
Schematic diagram of sensor packaging.

**Figure 13 micromachines-14-01657-f013:**
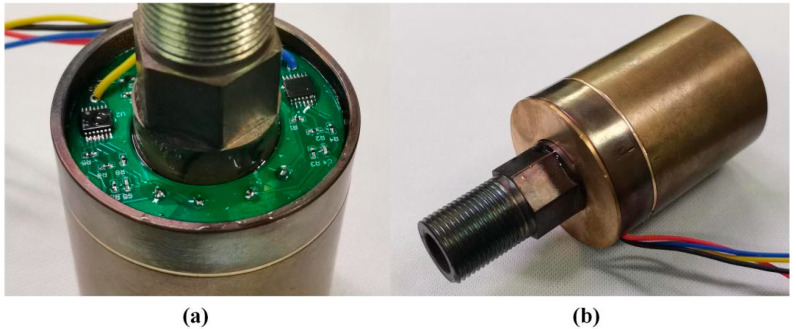
Physical image of sensor prototype: (**a**) Circuit lower chamber; (**b**) Cap packaging.

**Figure 14 micromachines-14-01657-f014:**
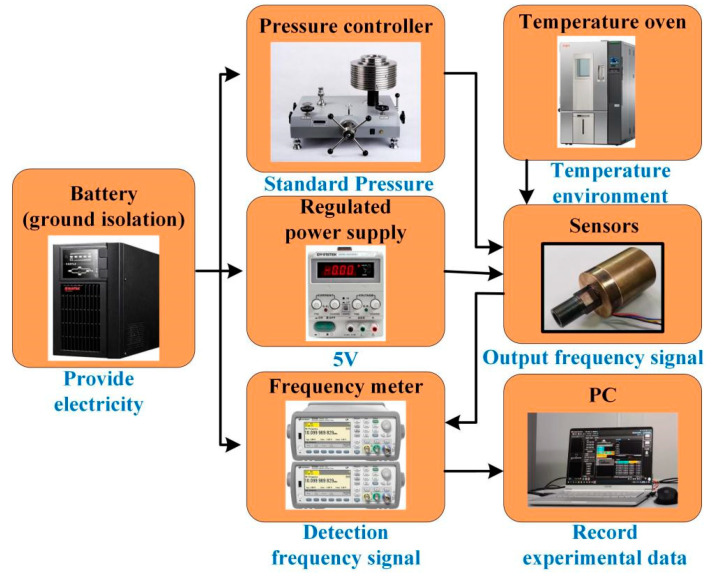
The equipment and experimental principle for performance experiment.

**Figure 15 micromachines-14-01657-f015:**
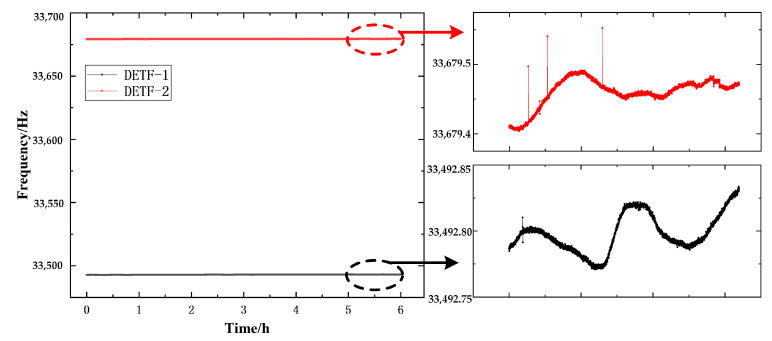
Zero time drift of two output frequencies of the resonant unit.

**Figure 16 micromachines-14-01657-f016:**
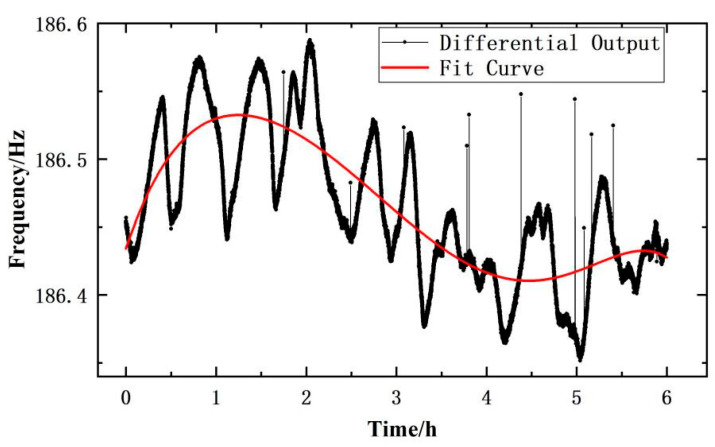
Zero time drift of differential output frequency of the sensor.

**Figure 17 micromachines-14-01657-f017:**
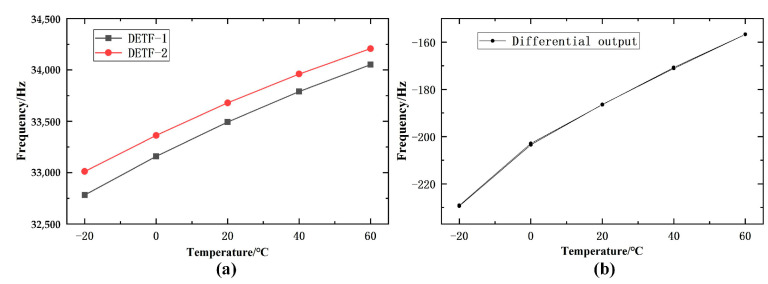
Zero time drift of the sensor: (**a**) Output of DETFs; (**b**) Differential output.

**Figure 18 micromachines-14-01657-f018:**
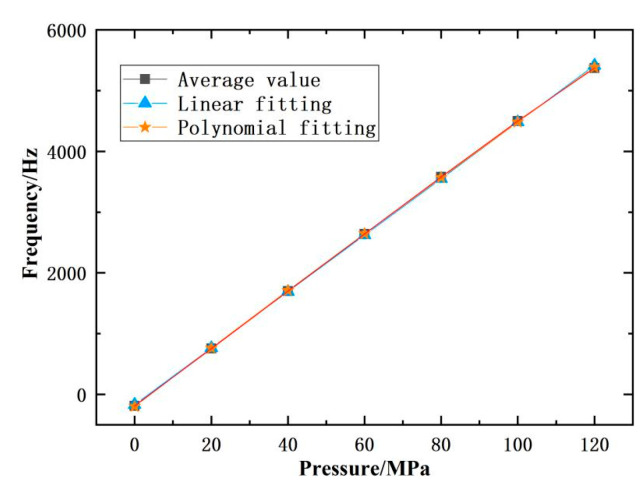
Experimental data on the comprehensive accuracy differential output of the sensor.

**Figure 19 micromachines-14-01657-f019:**
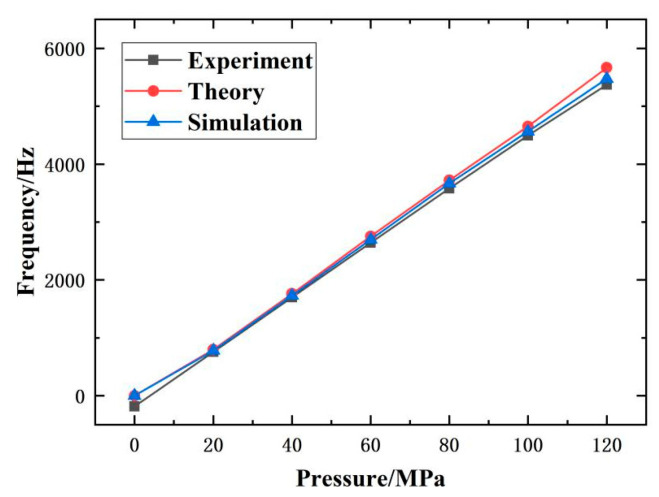
Comparison of experimental results with theoretical and simulation analysis.

**Figure 20 micromachines-14-01657-f020:**
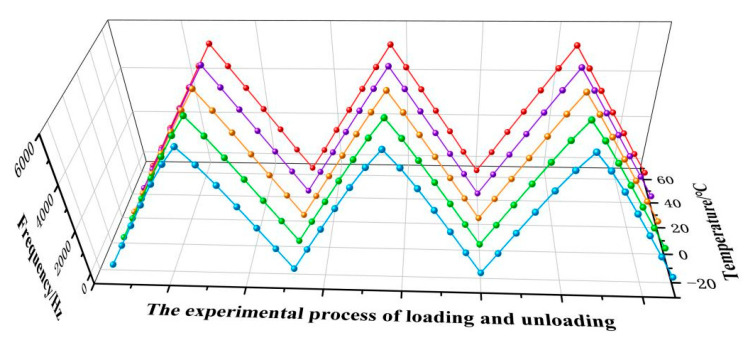
Experimental results of static calibration in full temperature range.

**Table 1 micromachines-14-01657-t001:** Structural parameters of the DETF.

Parameters	Value
Length of the tuning fork tines	4800 µm
Width of the tuning fork tines	145 µm
Thickness of the tuning fork tines	100 µm
Total length of DETF	9600 µm
Theoretical working frequency	34 kHz
Simulation working frequency	35 kHz
Actual working frequency	33.6 kHz

**Table 2 micromachines-14-01657-t002:** The performance of the sensor prototype in the full temperature range environment.

	−20 °C	0 °C	20 °C	40 °C	60 °C
Zero position/Hz	−229.95	−203.92	−186.99	−171.88	−158.39
Full-Scale output/Hz	5289.18	5560.08	5558.01	5600.87	5611.85
Sensitivity/(Hz/MPa)	44.08	46.33	46.312	46.67	46.77
Fitting error/% FS	0.025	0.026	0.018	0.020	0.023
Hysteresis error/% FS	0.011	0.01	0.011	0.011	0.015
Repeatability error/% FS	0.0083	0.0063	0.0079	0.0073	0.0086
Comprehensive accuracy/% FS	0.0285	0.0286	0.0224	0.0235	0.0288

## Data Availability

The data supporting the results of this study can be obtained from the first author according to reasonable requirements.

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
