# Peer review of "A High-Precision Quartz Resonant Ultra-High Pressure Sensor with Integrated Pressure Conversion Structure"

_micromachines, 2023, doi:10.3390/mi14091657_

Round 1
Reviewer 1 Report
In section 2,the author should cite EERNISSE patent that is the first to present the DETF force sensor:
"Eer Nisse, E. P. (1980). Miniature quartz resonator force transducer (No. US 4215570; A)."
The author should specify what type of simulation he is implementing.
The theoretical analysis of the resonator Q factor could also be presented.
In line 283, the sentence is not finished: "the surface electrode has been made by sputtering and inclined evapora-282 tion and [15] "
In section 3, the pattering of side electrodes could be detailed.
Author Response
Responses to Reviewer 1:
Sincere thanks for your serious and careful reviewing of our manuscript. Following is our response to your comments:
- Comment: In section 2,the author should cite EERNISSE patent that is the first to present the DETF force sensor: "Eer Nisse, E. P. (1980). Miniature quartz resonator force transducer (No. US 4215570; A)."
Response: Thank you very much for your precious time and careful reviewing of our manuscript, and sincere thanks again for giving our paper a positive consideration, as well as the important guiding significance to our research.
Sorry for not citing EERNISSE's patent. At the beginning of section 2, this paper added a reference to this patent. This is of great help to the rigor of the cited literature in this paper.
The specific modifications in the manuscript are as follows:
Page 3, Line 104: This paper added a reference ([14]) to this patent.
Page 20, Line 635: ‘14. Nisse E P E .Miniature quartz resonator force transducer [P].US 4215570A.’ is added.
- Comment: The author should specify what type of simulation he is implementing.
Response: Thank you very much for your suggestion. Sorry for the insufficient description of the simulation.
In order to further verify the accuracy of theoretical analysis and make up for any omissions in theoretical analysis, COMSOL finite element calculation software has been used for simulation verification in this paper. The modal simulation module has been used for the analysis of DETF, specifically simulating the working modal frequency of DETF and the force frequency sensitivity coefficient under different size parameters; The analysis of the sensor structure design used a solid mechanics module, mainly simulating the force acting on the sensor pressure conversion unit and resonance unit under the pressure to be measured.
The specific modifications in the manuscript are as follows:
Page 4, Line 140~142: “In order to verify the above theoretical analysis and further optimize the characteristic size parameters of the DETF, simulation optimization has been carried out.” is corrected to “In order to verify the above theoretical analysis and further optimize the characteristic size parameters of the DETF, the COMSOL (calculation software of finite element) has been used for simulation optimization.”
Page 4, Line 146~148: ‘The modal simulation module has been used to simulate the working modal frequency and force-frequency sensitivity coefficient of DETF under different size parameters.’ is added.
Page 9, Line 272~274: ‘The analysis of the sensor structure design uses the solid mechanics module of COMSOL, mainly simulating the force acting on the sensor pressure conversion unit and resonant unit under the pressure to be measured.’ is added.
- Comment: The theoretical analysis of the resonator Q factor could also be presented.
Response: Thank you very much for your suggestion. Sorry for not including a theoretical analysis of the resonator Q factor in this paper.
Our team described their research on the quality factor of DETF in another paper, and established an optimization model for force frequency characteristics and quality factor based on the vibration theory and damping mechanism of resonators. Due to the limited space of this paper, the initial draft did not include any discussion in this regard. Revise here and add ideas and results for the theoretical analysis of the quality factor of DETF.
The specific modifications in the manuscript are as follows:
Page 3~4, Line 129~138: ‘The quality factor is one of the important indicators of the resonant pressure sensor design scheme in this paper, which mainly affects the energy loss, frequency stability, and frequency selection ability of the resonant unit during vibration. The quality factor of res-onator components is related to their geometric structure parameters, gas packaging en-vironment, and other factors, mainly depending on various energy loss mechanisms. The main factors that affect the quality factor of quartz DETF are structural damping and gas damping. The structural damping losses of DETF resonators include thermoelastic losses, support losses, and surface energy losses. The gas damping of DETF mainly includes the air resistance damping of the resonant device, the sliding film damping of the upper and lower surfaces, and the squeeze film damping between the two tines [16].’ is added.
Page 20, Line 638~639: ‘16. Quanwei Zhang , Cun Li , Yulong Zhao , et al. A quartz resonant ultra-high pressure sensor with high precision and high stability[J]. IEEE Sensors Journal, 2021, 21(20): 22553-22561.’ is added.
- Comment: In line 283, the sentence is not finished: "the surface electrode has been made by sputtering and inclined evapora-282 tion and [15] "
Response: Sorry for the writing error, and this error has been corrected.
The specific modifications in the manuscript are as follows:
Page 11, Line 325~328: “In summary, the shape of the quartz DETF resonator structure has been completed by wet etching, the surface electrode has been made by sputtering and inclined evaporation and [15].” is corrected to “In summary, the shape of the quartz DETF resonator structure has been completed through wet etching, and the surface electrodes are made by sputtering [17].”
- Comment: In section 3, the pattering of side electrodes could be detailed.
Response: Thank you very much for your suggestion. Sorry for not providing a detailed description of the production of the side electrode.
In order to solve the problem of making side electrodes, this paper adopts a process plan that combines mechanical hard mask and inclined sputtering. Using magnetron sputtering equipment, the excited metal atoms move in a straight line in a high vacuum sputtering chamber, tilting the slide table at a certain angle to allow the metal atoms to pass through the hollow area of the mechanical hard mask and deposit on the side of the DETF to form a specific pattern.
The specific modifications in the manuscript are as follows:
Page 11, Line 329~334: ‘In order to solve the production problem of side electrodes, a process scheme combining mechanical hard mask and inclined sputtering has been adopted. Using magnetron sputtering equipment, the excited metal atoms move in a straight line in a high vacuum sputtering chamber, tilting the slide table at a certain angle to allow the metal atoms to pass through the hollow area of the mechanical hard mask and deposit on the side of the DETF to form a specific pattern.’ is added.

Reviewer 2 Report
The paper presents a detailed study on a push-pull integrated structure involving pressure circular membranes, square rods, flexible push rods, and resonant units. The research focuses on the design, theoretical analysis, simulation, and optimization of a sensor system that can convert pressure into mechanical signals. The system's operating mechanism is explored, and the paper provides insights into the deformation, stress distribution, and overall performance of the sensor.
The authors propose a push-pull differential structure to improve the measurement accuracy of the sensor. The theoretical model is established, and simulation analysis is conducted to optimize the structural size parameters. The paper also compares the theoretical values with simulation results, verifying the effectiveness of the proposed scheme.
Specific Questions and Comments
- Line 152-173: The introduction of the push-pull integrated structure is well explained. However, it would be beneficial to include more background information on the existing challenges that this structure aims to overcome. What are the specific advantages of this design over traditional methods?
- Line 174-210: The theoretical analysis section provides equations and descriptions of the mechanical equations of the structure. Could the authors elaborate on the choice of materials (e.g., beryllium copper alloy) and how they impact the overall performance of the sensor?
- Line 188-245: The simulation analysis section is comprehensive but might benefit from a more detailed explanation of the methodology used for simulation. What software or tools were used, and how were the parameters chosen for the simulation?
- Line 255-277: The results section compares theoretical calculations with simulation verification. While the conclusion mentions a small error, it would be helpful to quantify this error and discuss its potential implications.
- Line 279: The paper seems to introduce a fabrication section, but the content is not provided in the available text. Will there be further details on the fabrication process, and how it aligns with the theoretical and simulation aspects?
- The paper refers to figures such as Figure 3, Figure 4, etc. It would be essential to ensure that these figures are clear, well-labeled, and provide valuable insights into the discussed concepts.
Author Response
Responses to Reviewer 2:
Special thanks to you for your precious time and valuable comments! Following is our response to your comments:
The paper presents a detailed study on a push-pull integrated structure involving pressure circular membranes, square rods, flexible push rods, and resonant units. The research focuses on the design, theoretical analysis, simulation, and optimization of a sensor system that can convert pressure into mechanical signals. The system's operating mechanism is explored, and the paper provides insights into the deformation, stress distribution, and overall performance of the sensor.
The authors propose a push-pull differential structure to improve the measurement accuracy of the sensor. The theoretical model is established, and simulation analysis is conducted to optimize the structural size parameters. The paper also compares the theoretical values with simulation results, verifying the effectiveness of the proposed scheme.
- Comment: Line 152-173: The introduction of the push-pull integrated structure is well explained. However, it would be beneficial to include more background information on the existing challenges that this structure aims to overcome. What are the specific advantages of this design over traditional methods?
Response: Thank you very much for your precious time and careful reviewing of our manuscript, and sincere thanks again for giving our paper a positive consideration, as well as the important guiding significance to our research. Sorry for not fully describing the specific advantages of this paper's design.
The sensor structure scheme proposed in this paper has the following advantages: the pressure conversion unit is composed of an integrated pressure circular film and a flexible push rod, avoiding the adverse effects of heterogeneous materials and split structures, and reducing the problem of random errors caused by residual stresses and small gaps during assembly and welding processes in the sensor structure; The resonant unit adopts a push-pull differential structure, which not only reduces conjugate interference, but also the DETF resonator is only subjected to axial force, avoiding the problem of uneven stress distribution on the two tuning fork tines caused by the bending moment in the width direction of the DETF, and improving the measurement accuracy of the sensor.
The specific modifications in the manuscript are as follows:
Page 5, Line 176~191: “A push-pull structure integrated sensor structure scheme has been proposed, where the pressure conversion unit is composed of a pressure circular membrane and a flexible push rod manufactured in one piece, avoiding the adverse effects of heterogeneous materials and split structures. In response to the problem of uneven stress distribution on the two tines caused by the width direction bending moment of the quartz DETF, the resonant unit of this scheme adopts a push-pull differential structure, and the quartz resonator is only subjected to axial force, improving the measurement accuracy of the sensor,” is corrected to “The pressure conversion unit is composed of an integrated pressure circular film and a flexible push rod, which avoids the adverse effects of heterogeneous materials and split structures, and reduces the problem of random errors caused by residual stresses and small gaps during assembly and welding processes in the sensor structure; The resonant unit adopts a push-pull differential structure, which not only reduces conjugate interference, but also the DETF resonator is only subjected to axial force, avoiding the problem of uneven stress distribution on the two tuning fork tines caused by the bending moment in the width direction of the DETF, and improving the measurement accuracy of the sensor.”
- Comment: Line 174-210: The theoretical analysis section provides equations and descriptions of the mechanical equations of the structure. Could the authors elaborate on the choice of materials (e.g., beryllium copper alloy) and how they impact the overall performance of the sensor?
Response: Thank you very much for your question. Sorry for not providing a sufficient description of the material selection for the sensor.
Quartz crystal has the advantages of excellent mechanical properties, good elasticity, high bending strength, minimal hysteresis and creep, high quality factor, and good frequency stability. It is an ideal material for making high-precision and highly stable resonant units. Moreover, quartz crystals also have excellent electrical properties, as their piezoelectric properties allow for simple excitation and detection of resonant units. Beryllium bronze has high hardness, elastic limit, fatigue limit, and wear resistance, as well as good corrosion resistance, thermal conductivity, conductivity, non magnetism, fatigue resistance, and stress relaxation resistance. It is widely used as a material for making elastic components in various important fields. Moreover, the difference in thermal expansion coefficients between beryllium copper alloy and quartz single crystal is relatively small, reducing the adverse effects of temperature environment. Considering the structure of the paper, move the modifications to the end of the introduction.
The specific modifications in the manuscript are as follows:
Page 2, Line 74~84: “Quartz has the advantages of excellent mechanical properties, good elasticity, high bending strength, minimal hysteresis and creep, high quality factor, and good frequency stability. It is an ideal material for making high-precision and highly stable resonant units. Moreover, quartz also has excellent electrical properties, as their piezoelectric properties allow for simple excitation and detection of resonant units. Beryllium bronze has high hardness, elastic limit, fatigue limit, and wear resistance, as well as good corrosion resistance, thermal conductivity, conductivity, non magnetism, fatigue resistance, and stress relaxation resistance. It is widely used as a material for making elastic components in various important fields. Moreover, the difference in thermal expansion coefficients be-tween beryllium copper alloy and quartz single crystal is relatively small, reducing the adverse effects of temperature environment.” is added.
- Comment: Line 188-245: The simulation analysis section is comprehensive but might benefit from a more detailed explanation of the methodology used for simulation. What software or tools were used, and how were the parameters chosen for the simulation?
Response: Thank you very much for your suggestion. Sorry for the insufficient description of the simulation.
In order to further verify the accuracy of theoretical analysis and make up for any omissions in theoretical analysis, COMSOL finite element calculation software was used for simulation verification in this paper. The modal simulation module was used for the analysis of DETF, specifically simulating the working modal frequency of DETF and the force frequency sensitivity coefficient under different size parameters; The analysis of the sensor structure design used a solid mechanics module, mainly simulating the force acting on the sensor pressure conversion unit and resonance unit under the pressure to be measured.
The specific modifications in the manuscript are as follows:
Page 4, Line 140~142: “In order to verify the above theoretical analysis and further optimize the characteristic size parameters of the DETF, simulation optimization has been carried out.” is corrected to “In order to verify the above theoretical analysis and further optimize the characteristic size parameters of the DETF, the COMSOL (calculation software of finite element) has been used for simulation optimization.”
Page 4, Line 146~148: ‘The modal simulation module has been used to simulate the working modal frequency and force-frequency sensitivity coefficient of DETF under different size parameters.’ is added.
Page 9, Line 272~274: ‘The analysis of the sensor structure design uses the solid mechanics module of COMSOL, mainly simulating the force acting on the sensor pressure conversion unit and resonant unit under the pressure to be measured.’ is added.
- Comment: Line 255-277: The results section compares theoretical calculations with simulation verification. While the conclusion mentions a small error, it would be helpful to quantify this error and discuss its potential implications.
Response: Thank you very much for your suggestion. Sorry for not fully describing the comparison between theoretical calculations and simulation validation.
In the theoretical and simulation analysis of the entire sensor in this paper, the two DETF resonators of the resonant unit are completely consistent. However, in the actual manufacturing process, there are certain differences in each DETF resonator, which affect its working mode frequency and force frequency sensitivity coefficient. For example, the zero output of both sensor theory and simulation analysis is 0, while the actual test zero of the sensor prototype is 186Hz. Therefore, this paper only compares the sensitivity of the sensor: the theoretical value is 48.09 Hz/MPa, the simulation value is 44.18 Hz/MPa, and the experimental value is 46.32 Hz/MPa. The relative errors between the theoretical and simulation results and the experimental results are 6.24% and 9.80%, respectively. Verify the feasibility of the sensor scheme in this paper, and prove the rationality of the theoretical analysis and simulation calculation in this paper.
- Comment: Line 279: The paper seems to introduce a fabrication section, but the content is not provided in the available text. Will there be further details on the fabrication process, and how it aligns with the theoretical and simulation aspects?
Response: Thank you very much for your suggestion. Sorry for not providing a sufficient description of the manufacturing section in section 3.
The production of MEMS quartz resonant pressure sensors in this paper mainly includes three aspects: firstly, the processing of resonant units, mainly including the processing technology of external dimensions and the production of surface electrodes; The second is the production of pressure conversion units, and a process plan is developed for the overall structure design of this paper; The third is the packaging scheme design of sensors. By controlling the dimensional accuracy and process flow, the consistency between the sensor prototype and the theoretical simulation model is achieved. For example, the key structures of the pressure conversion unit (push rod, pressure film, and flexible rod, etc.) adopt the process of electric spark wire cutting and etching to ensure dimensional accuracy.
The specific modifications in the manuscript are as follows:
Page 10, Line 319~323: ‘The production of MEMS quartz resonant pressure sensors in this paper mainly includes three aspects: firstly, the processing of resonant units, mainly including external dimension processing technology and surface electrode production; The second is the production of pressure conversion units, and a process plan is developed for the overall structure design of this paper; The third is the packaging scheme design of sensors.’ is added.
- Comment: The paper refers to figures such as Figure 3, Figure 4, etc. It would be essential to ensure that these figures are clear, well-labeled, and provide valuable insights into the discussed concepts.
Response: Thank you very much for your suggestion. Sorry for the unclear drawing and labeling of these figures. Modifications have been made to Figure 3 and Figure 4 that are not clear enough and have issues with labeling.
The specific modifications in the manuscript are as follows:
Page 6, Line 192: Figure 3 is corrected.
Page 7, Line 203: Figure 4 is corrected.
